# Fiber-Optic Hydrophone Based on Michelson’s Interferometer with Active Stabilization for Liquid Volume Measurement

**DOI:** 10.3390/s22124404

**Published:** 2022-06-10

**Authors:** Welton Sthel Duque, Camilo Arturo Rodríguez Díaz, Arnaldo Gomes Leal-Junior, Anselmo Frizera

**Affiliations:** Telecommunications Laboratory (LABTEL), Graduate Program in Electrical Engineering, Federal University of Espírito Santo (UFES), Vitória 29075-910, ES, Brazil; wduque@gmail.com (W.S.D.); leal-junior.arnaldo@ieee.org (A.G.L.-J.); frizera@ieee.org (A.F.)

**Keywords:** liquid volume measurement, fiber-optic hydrophone, Michelson’s interferometer, ultrasound acoustics, active stabilization, machine learning

## Abstract

Sensing technologies using optical fibers have been studied and applied since the 1970s in oil and gas, industrial, medical, aerospace, and civil areas. Detecting ultrasound acoustic waves through fiber-optic hydrophone (FOH) sensors can be one solution for continuous measurement of volumes inside production tanks used by these industries. This work presents an FOH system composed of two optical fiber coils made with commercial single mode fiber (SMF) working in the sensor head of a Michelson’s interferometer (MI) supported by an active stabilization mechanism that drives another optical coil wound around a piezoelectric actuator (PZT) in the reference arm to mitigate external mechanical and thermal noise from the environment. A 1000 mL glass graduated cylinder filled with water is used as a test tank, inside which the sensor head and an ultrasound source are placed. For detection, amplitudes and phases are measured, and machine learning algorithms predict their respective liquid volumes. The acoustic waves create patterns electronically detected with resolution of 1 mL and sensitivity of 340 mrad/mL and 70 mvolts/mL. The nonlinear behavior of both measurands requires classification, distance metrics, and regression algorithms to define an adequate model. The results show the system can determine liquid volumes with an accuracy of 99.4% using a k-nearest neighbors (k-NN) classification with one neighbor and Manhattan’s distance. Moreover, Gaussian process regression using rational quadratic metrics presented a root mean squared error (RMSE) of 0.211 mL.

## 1. Introduction

Optical fibers have been used worldwide over long distances for ultra-high bitrates in telecommunication systems [1] and for sensor applications [2], including sensing acoustic waves through light modulation [3]. The use of optical fibers as acoustic sensors was first demonstrated in 1977 [4]. Optical fiber sensors (OFSs) have been studied and applied in the industries of petroleum exploration, fuel storage and transportation, industrial manufacturing, military, chemical processing, medical, aerospace, food and civil engineering [5,6,7].

The oil and gas industry requires control techniques for their vessels, with the aim of increasing production rates, avoiding environmental contamination, mitigating human labor risks, and reducing the costs of plant setup investments (CAPEX), operations, and maintenance (OPEX) [8]. A problem in these industries is the need for continuous monitoring of liquid volumes inside separation vessels working under the presence of dynamic emulsion and foam layers, accumulation of unwanted solids, sand in the bottom and sludge on the walls, turbulence caused by fluid injections, corrosive substances, and explosive atmospheres [9,10].

Ultrasound-based sensors employ acoustic transmitters and receivers immersed in liquids to calculate the acoustic impedance between adjacent layers of fluids [11]. For reception, piezoelectric hydrophones can be constructed with polymers, such as polyvinyledine fluoride (PVDF), that enable them to provide an output voltage that varies according to acoustic pressure [12]. Besides PVDF, there are ultrasound sensors made with optical fibers [13,14,15,16,17,18,19,20,21,22].

Compared to other sensing technologies, optical fiber sensors (OFSs) offer advantages as they are electrically (galvanically) isolated, immune to electromagnetic interference (EMI), intrinsically safe, resistant to chemical corrosion, useable at elevated temperatures, have a wide bandwidth, are capable of multiplexing, are minimally invasive, have good accuracy and resolution, have reduced sizes and weights, their interrogator systems can be installed far from the remotely monitored points, and they do not require any electrical power at the measuring points [8,23,24]. OFSs are precise in measuring pressure, temperature, acoustic fields, strain, torsion, deformation, curvature, force, vibration, acceleration, rotation, humidity, viscosity, and chemical parameters [7,25,26].

In 1977, the authors of [4] first demonstrated the possibility of a direct acousto–optic interaction between an ultrasound field and an optical fiber coil working as an acoustic sensor. The acousto–optic effect is based on the physics principle that a propagating acoustic wave modifies the refractive index of a fluid through pressure variations and densities [20] as a function of time and space. The authors of [4] used 4 m of an SMF optical fiber section length to build a coil of 10 turns and 3.3 cm diameter and submerged it into water so that its optical beam was phase modulated by ultrasound waves ranging from 40 to 400 kHz. The optical fiber coil sensitivity was characterized by light phase shifts whose intensity depended on the acoustic pressure incident on the optical fiber length, with which an MI has been employed [27,28,29].

A fiber-optic hydrophone (FOH) is an acoustic sensor that uses optical fibers as the sensing element, and can be applied in fields such as oil and gas exploration, earthquake inspection, and underwater object detection [15,30]. FOH is typically implemented based on optical interferometry, which offers greater sensitivity than piezoelectric hydrophones, and they are classed as phase-modulated sensors [7,31,32]. An FOH system measures the light phase changes induced by a particular measurand [7], in which an ultrasound piezoelectric source (UT) works as the transmitter, and an optical fiber sensor works as the receiver [11,33].

Interferometric schemes can be mounted by using commercially available optical components [34]. Four interferometer configurations are usually cited in the literature: Fabry–Perot, Mach–Zehnder, Michelson’s, and Sagnac, as they provide good sensitivity, accuracy, large dynamic ranges, and may cover long distances of monitored points [32]. The latter three types are also known as two-beam interferometers [35].

Publications about FOHs have demonstrated their working principles, showing them as an alternative technology to replace piezoelectric hydrophones [36]. FOHs do not need recalibration [37] and present higher bandwidths than conventional piezoelectric hydrophones [38]. Interferometric techniques can be used to improve their sensitivity [39], resolution, bandwidth, dynamic range, and signal-to-noise ratio [40]. A twisted pair made with single-mode fibers has been recently published as a high-sensitivity, broadband ultrasound sensor able to work from 20 kHz to 94.4 MHz [41]. These publications are focused on the technological aspects of FOH.

Other papers present applications and proof of concepts using FOH sensors. As examples, the largest array of FOHs in the world (16,000 sensor elements) is used undersea to permanently monitor oil reservoirs in the North Sea [15], an FOH is used to precisely detect cavitation bubbles under high ultrasound emissions [42], two 32-element line arrays were deployed off the San Diego’s coast for object recognition underseas [43], an FOH was used to measure the power of hyperthermia transducers without being damaged [44], and a 50 MHz wideband FOH was developed to measure medical ultrasound fields [45].

To the best of our knowledge, there are no published works describing laboratory prototypes of an FOH system to measure liquid volumes and presenting results of accuracies and RMSE derived from machine-learning algorithms. There are publications describing the behavior of underwater optical coils, but they do not present results of using these coils to measure liquid volumes [16,17,18,19,46,47,48,49]. Ultrasound methods to determine multilayers of oil, emulsion, and water in oil tanks are demonstrated in [10,46,50] and show the advantages of FOH, such as contactless distance measurement, low costs, high precision, simple setup, and imperviousness to dusty and smoky environment. However, there were not results regarding liquid volume measurements.

Considering an FOH system as a new generation [15] and a primary application [40] for underwater acoustic sensors and underwater detection [19], the objective of this work is to present a prototype composed of optical fiber coils in the MI’s sensor head, with an active stabilization mechanism driving another optical coil wound around a piezoelectric actuator in the MI’s reference arm, and a software application that uses amplitudes and phases of detected acoustic signals to predict liquid volume by machine learning algorithms.

Although the isolated functions of the purposed system have been already studied and applied in known applications, the novelty of this proposal is the bringing together of those functions, such as: underwater acoustic wave detection; optical coils as sensing elements; an optical interferometry scheme; an electronic circuit for homodyne detection and active stabilization of noise; another optical coil wound around a piezoelectric actuator; the use of electric phase differences between the acoustic signals inputted and outputted by the sensor; and the definition, testing and comparison of several data models processed by machine learning algorithms, offering new insights and results with high accuracies and low errors for liquid volume measurement.

Section 1 introduces fiber-optic sensors, hydrophones, and related applications. Section 2 presents a theoretical background of acoustic detection, and Section 3 describes the materials and methods employed in the experimental setup. Then, Section 4 presents the results and discussion. Finally, Section 5 ends the paper with conclusions and future work suggestions.

## 2. Theoretical Background

The MI has been chosen for this work due to its simple configuration and the need for fewer fiber splices and one beam splitter [32,40]. Besides, it accepts flexible geometries and has lower component costs compared to other schemes [7,51]. These characteristics were taken into consideration to choose the MI as the sensor head for this work.

### 2.1. Michelson’s Interferometer

An MI, whose picture is adapted from [7] (p. 371), is presented in Figure 1, in which a 3 dB beam splitter (BS) divides the power of an input coherent laser [31] between two arms defined as optical paths, which end in mirrors that reflect the optical beams back. The optical paths can be free-space air, tubes with gases, vacuum, or optical fibers. The two beams are then recombined by the BS, and a photodetector outputs the phase changes from the interferometric resulting signal.

Considering an MI fed with power $P_{\mathrm{IN}}$, the light wavelength in the medium λ, a wave number k=2π/λ, and a lossless beam splitter with a division rate of 0.5, the relation between the output power $P_{\mathrm{OUT}}$ and the interferometer’s optical path differences $\Delta L=L_{S}-L_{R}$ can be expressed by [35]:(1)$$P_{\mathrm{OUT}}=\frac{P_{\mathrm{IN}}}{2}\left[1+\cos\left(\frac{4\pi \Delta L}{\lambda }\right)\right]$$

The first MI arm is called the “sensor head” or “wet arm”; it has a length $L_{S}$ and is exposed to measurands, such as sound, pressure, temperature, strain, and mechanical vibrations. The other arm is known as the “reference arm” or “dry arm”; it has a length $L_{R}$ and should be isolated from environment disturbances [7].

A desired measurand induces changes on the sensor head that dynamically change its physical fiber length $L_{S}$, leading to a path difference ΔL among the two arms that induces phase differences Δϕ between their light beams, resulting in phase variations that are electronically expressed by a photodetector [7].

Linear polarization of a light beam in one MI arm can be expressed by [31]:(2)$$\vec{E}\left(z,t\right)=E_{0}\cos\left[\omega_{0}t-\left(\frac{2\pi }{\lambda }\right)z\right],$$
where $\vec{E}$ is the electrical field of the optical electromagnetic wave, $E_{0}$ is its maximum amplitude, $\omega_{0}$ is the optical wave frequency in radians per seconds, z is the longitudinal space point, and t is time.

Equation (2) shows amplitudes that might be present on the transverse $\vec{x}$ or on the $\vec{y}$ space coordinates, propagating through the longitudinal $\vec{z}$ coordinate. Considering the fiber’s physical length L, the light speed in vacuum $c_{0}$, the vacuum wavelength $\lambda_{0}$, the fiber refraction index n, the wave group velocity $v=c_{0}/n$, which leads to $1/\lambda =n/\lambda_{0}$, the electrical field can be expressed by:(3)$$\vec{E}=E_{0}\cos\left[\omega_{0}t-\left(\frac{2\pi n}{\lambda_{0}}\right)L\right],$$
and Equation (4) shows the fixed phase angle $\phi_{0}$ (in radians) of the light beam. The term “nL” is defined as the “optical path length” [52].
(4)$$\phi_{0}=\frac{2\pi \mathrm{nL}}{\lambda_{0}},$$
and changes in L or n lead to phase variations, expressed by:(5)$$\phi_{0}+\Delta \phi =\frac{2\pi }{\lambda_{0}}\left[n\cdot L+n\cdot \Delta L+\Delta n\cdot L\right],$$
where Δϕ are the incremental phase changes due to incremental length changes ΔL, which induce refraction index variations Δn due to the optical fiber’s photo–elastic effect. Further, frequency variations (jitter) from the laser source may also contribute to phase drifts, and Equation (5) can be expressed as a sum of derivatives of the terms involved in the phase drifts [52]:(6)$$\frac{d\phi }{\phi }=\frac{\mathrm{dL}}{L}+\frac{\mathrm{dn}}{n}+\frac{\mathrm{dk}}{k}$$

Current optical detection systems estimate the average power per unit area and unit time as a measure of intensity or irradiance $I_{\mathrm{OUT}}$, which is proportional to the squared amplitude of the electrical field [53]:(7)$$I_{\mathrm{OUT}}\approx E_{0}{}^{2}$$

The summation of both arms’ light waves (${\vec{E}}_{T}={\vec{E}}_{R}+{\vec{E}}_{s}$) results in an irradiance perceived by the photodetector [31] that can be expressed by:(8)$$I_{\mathrm{OUT}}=2E_{0}{}^{2}\left[1+\cos\left(\Delta \phi \right)\right],$$
where $\Delta \phi =\phi_{R}-\phi_{S}$ is the phase difference of both light waves’ summation.

In another view, this phase difference can also be represented by $\Delta \phi =\phi_{e}-d\phi_{m}$, in which $\phi_{e}$ is the bias phase standing for the external disturbances with low frequency and slow drifts with time, affecting both interferometer arms (sensor and reference). The second term $d\phi_{m}$ represents the pure physical measurand variations of interest, whose value represents the interferometer’s sensor arm [40].

When the interferometer is forced to keep its bias phase $\phi_{e}$ around (2n+1)∗π/2 radians points, it reaches a condition called quadrature state, in which it supplies a better response to the measurand $d\phi_{m}$. The nature of sinusoidal waves at the quadrature point limits the two-beam interferometers to work under the largest optical phase displacements of half a wavelength. Moreover, when the bias phase $\phi_{e}$ is located around zero or on multiple integers of ±π radians, the resulting interferometric pattern jumps between zero and maximum peak values, representing the worst sensitivity points, as the derivatives of intensity to phase are zero at them [31,35,40,54].

Although Δϕ may appear to be a stable measurand, the noise present on $\phi_{e}$ can be so elevated that the measurand $d\phi_{M}$ may become impossible to read [31,40]. More details about the MI working as an optical hydrophone for signal intensity (irradiance) and phase (homodyne or heterodyne) measurand detection can be obtained from other studies: [7,15,18,19,25,31,32,40,52,54,55,56].

### 2.2. Michelson’s Interferometer for Acoustic Wave Detection

Taking into consideration the equations and terms already presented, an MI whose sensor arm is composed of a single-mode optical fiber coil immersed in liquid, within which propagates an acoustic field of pressure $P_{a}$ and angular frequency $\omega_{a}$, the interferometer’s resulting light beam is expressed by [4]:(9)$$\vec{E}=E_{0}\exp i\left(\omega_{o}t\right)+E_{0}\exp i\left(\omega_{o}t+K_{a}\sin\left(\omega_{a}t\right)+\Delta \phi \right),$$
where $K_{a}$ is the modulation index that defines the acoustic influence on the phase drifts, and Δϕ are the dynamic optical phase differences between both MI arms. For a uniform acoustic field, $K_{a}$ is expressed below, and $L_{c}$ is the physical length of the optical fiber coil interacting with the acoustic wave [4]:(10)$$K_{a}=\frac{2\pi }{\lambda_{0}}\frac{\partial n}{\partial P_{a}}P_{a}{\mathrm{nL}}_{c}$$

Bucaro et al. [4] also define $S_{d}$ as constant that defines the detector’s sensitivity, which depends on the photodetector’s gain G, quantum efficiency q, electron charge $e=1.602\times 10^{-19}\;c$, Planck’s constant $h=6.626\times 10^{-34}{\;m}^{2}\mathrm{kg}/s$, and on other terms already presented, as expressed by:(11)$$S_{d}=\frac{2\pi \mathrm{Gqe}}{{\mathrm{hc}}_{0}k}$$

Finally, [4,57] have shown that small values of modulation index $K_{a}$ will result in the following irradiance equation as perceived by the MI hydrophone:(12)$$I_{\mathrm{OUT}}=4E_{0}{}^{2}S_{d}\left[1+\cos\left(\Delta \phi \right)-\frac{K_{a}}{2}\sin\left(\Delta \phi \right)\sin\left(\omega_{a}t\right)\right]$$

### 2.3. Active Stabilization for Homodyne Detection

The phase instabilities inherent to optical interferometers represent a challenge that requires countermeasures to mitigate [35]. In an ideal scenario, the MI works properly when its arms are equal in length (balanced condition) and the reference arm is isolated from external noise [18,54,55,56]. Thus, MI measurement noise needs to be controlled by adjustment mechanisms [6,7,8], such as Pockels cells [58], rubber and springs for anti-vibration [28], optical tables, hermetic boxes, thermal baths [56], piezoelectric actuators [18,55], sensing information encoded in a carrier signal, phase generated carriers, and active homodyne demodulation [40]. Despite the solutions aiming to achieve MI stabilization, they are not always continuously stable over time, and, as reported by [28], sometimes noise isolation is “still an art rather than a science”.

## 3. Materials and Methods

This system works based on physics principles in which acoustic wave behavior depends on liquid density, reflections from internal tank walls and other components, and on changes to the acoustic path characterized by changes in the volume of liquid [46].

### 3.1. Experimental Setup

The experiment setup in Figure 2 shows an FOH system for liquid volume measurement, based on the MI, with an electronic feedback (EF) circuit loop to compensate for external noise from the sensor’s measurands. The system outputs a sinusoidal signal from which two values are extracted: amplitude and phase. The phase is the difference between the acoustic field delivered to the UT and the acoustic field detected by two optical coils S1 and S2 placed at the sensor head. For each differential volume of water added to or extracted from the tank, the length of the water (acoustic path) changes, and the internal water surface dislocates. Thus, a new profile of internal sound reflections and backscattering inside the tank creates an interferometric and stationary pattern, knowing that internal water interface works as a mirror to the acoustic waves. Then, the optical coils capture the ultrasound intensities present at their fixed positions, modulating light beams through their photo–elastic (acousto–optic) effects, creating an output signal from which the amplitudes and phases are extracted.

It is not trivial to mathematically equate the components of this setup to reach an expression that infers the water’s liquid volume by using the system’s output. This setup involves optical and acoustic phenomena taking place simultaneously by an unbalanced MI composed of two optical coils in the same sensor arm with noise stabilized by an EF circuit driving a piezoelectric actuator in the reference arm. The coils are not multiplexed.

### 3.2. Hardware Components

The setup worked for seven days in an air-conditioned room and water temperatures varying from 22 °C to 24 °C. The test tank was a 1000 mL glass graduated measuring cylinder with 10 mL grading divisions filled with water one milliliter at a time from 440 mL to 1000 mL (561 liquid volume points) with a syringe graduated at 1 mL. The measurements started at 440 mL due to setup arrangements and positioning of the UT inside and in the bottom of the tank. The irradiating face of the UT was positioned at 300 mL volume due to its mechanical size, and the distance of 140 mL between the UT’s face and first coil S1 was due to the near field of Fresnel’s zone.

A coherent laser light source TN Laser (Teraxion Narrow Linewidth Laser) was tuned at 1550 nm, and its wavelength drift was negligible as it implements internal hardware for electronic current and temperature stabilization. The TNL laser injected 12.5 dBm of light power in the system. The isolator IS (Thorlabs, Newton, NJ, USA) protected the laser source port against undesired returning power. After passing through circulator C1, polarization controller PC1 and the 3 dB beam splitter BS, the light power was equally injected in both MI arms, which ended in Faraday-rotator mirrors M1 and M2. The MI’s reference arm was composed of an optical coil of 11 m [55] made with SMF fiber (Corning SMF-125/9, only with acrylate protection) wound around a piezoelectric actuator PZT (Thorlabs APF705) responsible for stabilization. The sensor arm’s coils S1 and S2 were both identical and made of the same fiber type, with an outer diameter of 2.4 cm, and 25 fiber turns, resulting in 188.5 cm of total fiber length per coil. In total, considering all fibers used, component connections, installation arrangements and coils, each MI arm achieved a length of around 15 m. Single mode fibers were used because the behavior and analysis of multimode interferometers are difficult to control [56].

All optical components, such as the isolator, circulators, beam splitter, Faraday mirrors and PZT with its optical coil, were installed inside boxes (non-hermetic) with the goal of mitigating external influences, such as temperature variations, random pressure fluctuations, sounds, air flows, or mechanical stress on fibers and connectors. The optical components and photodetectors were mounted over an optical bread board with a rubber foot to mitigate mechanical vibration and vibration on the table where the components were situated. These disturbances were controlled as they could decrease measurand accuracy [18,55,56].

The polarization controller PC1 was placed at the MI input to adjust the polarization and improve the contrast [56]. Following the same idea, PC2 was installed at the MI output, which delivers the sensor’s measurand signal to the photodetector PD3. The two circulators C1 and C2 provided the required optical paths [55] to photodetectors PD1, PD2, and PD3 (Thorlabs PDA30B-EC). This optical arrangement guaranteed that the optical signals input to PD1 and PD2 were 180° out of phase. While both photodetectors PD1 and PD2 were only working for the stabilization mechanism, PD3 was uniquely responsible for supplying the sensor system’s output signal. The EF circuit was fed by PD1 and PD2, and its output drove the PZT, which stretched the optical coil of the MI’s reference arm, keeping the interferometer locked in quadrature state and providing a final sensing signal with less interference through PD3.

On the electronic side, the Signal Generator SG (BK Precision 4053) was set to provide a 100 kHz continuous sinusoidal wave (CW) on both its output channels, CH1 and CH2, with the same phases but different peak-to-peak voltages of 10 Vpp and 1.8 Vpp, respectively. During the preliminary tests, a range of UT frequencies were evaluated from 50 kHz to 1 MHz, and an adequate response range of voltage and phase variations occurred at 100 kHz. The lower UT (Precision Acoustics Unfocussed 1 MHz, 23 mm diameter and 75 mm focal distance) frequency was used to create higher wavelength sounds in the water, supplying a better response to the volume variations of 1 mL.

The SG-CH1 signal was amplified by the Juntek DPA-2698 (BW 10 MHz and 3 dB gain) amplifier (A), which output a 20 Vpp sinusoidal CW signal of 5 W to the UT. The second port of signal generator SG-CH2 was connected to the CH1 input of the Oscilloscope OP (Tektronix Open Choice MDO 3012) to work as a reference for FOH system phase recovery. The UT was positioned in such a way that the ultrasound waves were irradiated upwards through the liquid, and their wavefronts were parallel to the plane of the optical coils S1 and S2. The FOH output was injected into the second oscilloscope port (OP-CH2).

Both OP ports (CH1 and CH2) were configured to supply AC coupling and BW of 20 MHz, making them more adequate for the involved frequencies and to cut DC power. The OP’s Fourier-transform function allowed seeing one channel in both time and frequency domains simultaneously, and the signal amplitude of 1.8 Vpp from SG-CH2 was chosen to help with adjustment during setup. Last, the MATLAB application (MA) [59] read the signals from both oscilloscope ports and extracted two measurands: amplitude in volts and phase in degrees. This phase value was calculated by MA and stood for the difference between the original signal injected into the UT (interferometer input) and the resulting signal delivered by PD3 (main MI output). This output was then digitalized by the oscilloscope at a sample rate of 100 MS/s, the samples were read in vectors of 10 thousand points of double type variables, which were passed through a software band-pass filter and then to a Fourier transformation, from which the values of amplitude and phase were registered in a database.

### 3.3. Fresnel’s Acoustic near Field

A Fresnel’s near field was kept free between the UT surface and the first coil S1. The acoustic’s near field was a Fresnel’s space that should not have been obstructed by any object so that the acoustic field [12] emitted by the UT would not be destroyed or severely attenuated, which would have compromised the acoustic detection by the optical coils in the MI’s sensor head. The calculation of this field considered the UT’s diameter (D=23 mm), the ultrasound frequency (f=100 kHz), and the wave speed in water [60]. At lower temperatures, the larger the near field; the worst case of T = 22 °C was used to calculate the water’s sound speed $s_{\mathrm{wa}}$ [60] and the near field distance $F_{\mathrm{nf}}$ [12] as follows:(13)$$s_{\mathrm{wa}}=1404.3+4.7\times T-0.04\times T^{2}=1488.34\;m/s$$

The near field for a cylindrical UT is expressed by:(14)$$F_{\mathrm{nf}}=\frac{{\left(\frac{D}{2}\right)}^{2}}{\lambda_{\mathrm{wa}}}=\frac{{\left(\frac{23\;\times \;10^{-3}}{2}\right)}^{2}}{\lambda_{\mathrm{wa}}}\;∴{\;\lambda }_{\mathrm{wa}}=\frac{s_{\mathrm{wa}}}{f}=\frac{1488.34}{100\times 10^{3}}\;∴{\;F}_{\mathrm{nf}}=8.9\;\mathrm{mm}$$

For the test tank of this experiment and $\lambda_{\mathrm{wa}}=14.9\;\left[\mathrm{mm}\right]$, the distance $F_{\mathrm{nf}}=8.9\;\left[\mathrm{mm}\right]$ is equivalent to the volume range $F_{\mathrm{nf}}=26.3\;\mathrm{mL}$. Considering that the UT surface was placed at the volume of 300 mL, the near field for f=100 kHz and T = 22 °C is found in the range from 300 mL to 326.3 mL and, therefore no objects, not even an optical coil sensor, should be put in this region.

As higher ultrasound frequencies were also used during the preliminary tests, the free water volume of 140 mL purposely left between the UT surface and the first S1 coil allowed the use of a maximum frequency of 520 kHz without obstructing the near field.

### 3.4. Electronic Feedback Loop Circuit

In this work, an active and self-compensate mechanism for the MI was adopted to control the environment disturbances and offer better stability to the measurand data perceived by the sensor head. It was based on the work developed by [18,55], who implemented the electronic feedback (EF) loop circuit shown in Figure 3.

The EF was composed of four operational amplifier chips (U1 to U4), a dual comparator chip U5, a dual flip-flop chip U6 (CD4013BC), one four-electronic keys chip U7 (CD4016BCN), and other discrete components such as resistors (18 units), capacitors (2 units), and diodes (3 units). This EF circuit has two inputs, whose signals are received from the photodetectors PD1 and PD2, and one signal output, which drives the PZT. The output values of PD1 and PD2 were matched through dB gain adjustment knobs of the photodetector’s hardware (Thorlabs PDA30B-EC).

Some minor changes were made to the original author’s electronics and optical schemes published by [55] to simplify the setup, but without affecting any original functionality of the circuit. Both input low pass filters were not electronically implemented, as they were provided by the oscilloscope input ports and by a software high-pass filter implemented in MATLAB. Furthermore, the DC power supplies were simplified to use the values of +10 and −10 volts, cutting the need for +5/−5 and +15/−15 VDC inputs. Finally, the four U7 electronic switches were connected in parallel to decrease their internal resistance and to allow proper PZT operation.

The active stabilization’s EF loop acts as a homodyne demodulator that nullifies as much as possible the output of the differential amplifier U3, locking the MI operation around the vicinities of the quadrature point [40]. The interferometric signals output by PD1 and PD2 (electronic currents) were 180° out of phase and input to the EF circuit for differential detection.
(15)$$I_{\mathrm{PD}1}=I_{0}\left[1+K_{0}\cos\left(\phi_{s}+\phi_{d}\right)\right],$$
(16)$$I_{\mathrm{PD}2}=I_{0}\left[1+K_{0}\cos\left(\phi_{s}+\phi_{d}+\pi \right)\right],$$
where $I_{0}$ relates to the system input power, $K_{0}$ is related to the interferometric fringe visibility, $\phi_{s}$ is the static differential phase, and $\phi_{d}$ is the disturbances’ differential phase.

After PD1 output is inverted by U1, and, together with PD2 output, both signals are differentially combined by the converter U2, the following signal is seen on the U2 output, where $K_{2}$ if the conversion gain of U2:(17)$$U2_{\mathrm{OUT}}=K_{2}\times \cos\left(\phi_{s}+\phi_{d}\right)$$

After $U2_{\mathrm{OUT}}$ is integrated by U3, the new output signal is presented below, where $K_{3}$ is the conversion gain of U3:(18)$$U3_{\mathrm{OUT}}=K_{3}\times \sin\left(\phi_{s}+\phi_{d}-\pi /2\right)$$

As the perfect quadrature point is defined by $\phi_{s}+\phi_{d}=\pi /2$, and the interferometer is forced to be kept around the quadrature state, then $U3_{\mathrm{OUT}}$ will tend to zero, knowing that, the differential phase will be:(19)$$\Delta \phi =\phi_{s}+\phi_{d}-\pi /2\;∴\;\Delta \phi ≅0$$

However, this absolute zero is not supported, and in the vicinity of the quadrature point, differential phase Δϕ drifts assume small values, allowing the approximation below, considering that sin(Δϕ)≅Δϕ for small values of the angle Δϕ:(20)$$U3_{\mathrm{OUT}}=K_{3}\times \sin\left(\Delta \phi \right)\;∴\;U3_{\mathrm{OUT}}≅K_{3}\times \Delta \phi ,$$

This simplification represents a practical assumption when a sine or cosine math function falls inside a cosine or sine formula, such as the term “$\exp i\left(\omega_{o}t+K_{a}\sin\left(\omega_{a}t\right)+\phi_{0}\right)$” in Equation (9). This assumption is considered valid for Δϕ≪1 and Δϕ≤0.1 radians as described as acceptable by [52]. In fact, it avoids the need to consider many terms of Bessel’s functions of the first kind, as the first two terms of their coefficient family, $J_{0}(\Delta \phi$) and $J_{1}\left(\Delta \phi \right)$, represent 98% of the homodyne power spectrum energy [52].

Therefore, $U3_{\mathrm{OUT}}$ is quasi-linear at the quadrature state [18]. It is used as the correction signal that drives the PZT, making displacements and strain changes in the optical fiber coil wound around it, keeping the interferometer working close to the quadrature state.

The EF circuit also tracks the external noise through an equivalent voltage in capacitor C1. The circuits U5.1 and U5.2 compare C1 with the reference voltage defined by the potentiometer POT (R18), and, when they are equal, the EF switches the C1 voltage to zero by an up-pulse to flip-flop U6 that keeps the four U7 switches closed and discharging by the time τ=R19×C2=0.47 ms. The reference value of +5 volts is selected with R18 (potentiometer) so that the limits of +5/−5 volts of accumulated phase drift errors are enough to protect the PZT and other electronic components, respecting their technical datasheets [18]. In general, the EF circuit can be regarded as a high-pass filter with a cutoff angular frequency $f_{\mathrm{EF}}$ in hertz, as expressed by [55]:(21)$$f_{\mathrm{EF}}=\frac{K_{p}K_{d}}{2\pi \times R8\times C1}$$

The voltage-phase coefficient $K_{p}$ rad/V for the PZT actuator and its optical fiber coil and the phase-voltage coefficient $K_{d}$ V/rad for photodetectors have been experimentally measured by driving the PZT with a low-frequency triangular wave [55], allowing calculation of $f_{\mathrm{EF}}$ Hz, and are further presented in the results section.

### 3.5. Sensitivity, Resolution, Bandwidth and Other Technical Characteristics

The purposed setup is composed of different elements, both optical and electric, each with specific technical parameters for power, frequency, bandwidth, sensitivity, resolution, internal noise sources, and others [40]. The proposed FOH system’s sensitivity, resolution, and bandwidth are dependent on the setup’s arrangement and on its components’ intrinsic characteristics, whose values are examined in the results section, following collected data series and their derivatives of phases and amplitudes.

Although it has not been the scope of this work to measure isolated parameters of the optical coils built for the MI’s sensor head, other studies have already demonstrated a sensitivity of −116 dB *re* rad/µPa in the bandwidth of 10 Hz to 2 kHz, phase noise of −102 dB *re* rad/$\sqrt{\mathrm{Hz}}$, and pressure noise of 14 dB *re* µPa/$\sqrt{\mathrm{Hz}}$ [15]. Another study shows a sensitivity of −170 dB *re* rad/µPa and noise floor of 50 dB *re* µPa/$\sqrt{\mathrm{Hz}}$ below 1 kHz [40]. These parameters vary with the type and length of fibers exposed to the acoustic waves, coil sizes and their layout, number of fiber layers, acoustic wave frequency and direction, mechanical and elasto–optic properties of fibers, and other parameters [51,61,62,63].

### 3.6. Data Collection and Database Structure

The MA developed for this experiment was connected online to all setup equipment and took twenty measurements of amplitude and phase per each milliliter of liquid volume, for a total of 11,220 readings (20 × 561) that were stored in a database. Values of amplitudes (volts) were rounded to 2 digits, and phases (degrees) were rounded to zero digits to the right of the decimal point to allow the system to calculate the modes (most frequent values) of both measurands.

Each reading was registered as a tuple of data object (lines) having 64 attribute fields (columns) of information. Among these fields, the tuple had point values of amplitude and phase, along with modes, means, standard deviations, and outlier indicators. A tuple also carried fields for both intermediate and final values of modes and means. Twenty measures composed a group, and the intermediate values were calculated based on the measurements taken until the moment of a point collection. For example, the 12th measurement of a group registered intermediate values of mean and mode for the twelve measures taken until that moment. The last measure of a group was calculated after removing the interquartile outliers. In this work, these outliers were values greater than 1.5 times the interquartile size, above the upper quartile (75%) and below the lower quartile (25%), for a measurement group. The interquartile method was applicable for data both normally distributed or not. The database also registered a field to show outliers that were less than 10 or above 90 percentile ranges. These two strategies are referred to in this text as “interquartile” and “percentile” outliers, respectively.

As temperature changes acoustic properties of liquids, each object tuple also had a field that registered water temperature, which was provided by an electronic sensor installed inside the test tank and connected to the MA. This field was used by the machine learning algorithms with the action of merely adding it or not among the columns of attributes selected to compose a data model to be assessed.

Further, no phase or amplitude values from repetitive cycles of increasing and decreasing volumes were registered in the database, despite the fact that this repeatability was seen during the setup tests.

### 3.7. Phase and Amplitude Data Series Analyses

This work presents data series analyses of amplitudes and phases, including their derivatives in relation to liquid volumes. Standard deviations and small linear intervals of these series were evaluated. All graphs show results before and after the S2 coil was submerged. The data series include the mode values of each measurand. First, the phase and amplitude were individually analyzed, and further both were put together in a two-dimensional Cartesian space. A specific phase analysis included three adjustment actions, after which an unwrapped phase characterization was obtained for the FOH system.

A limitation in the arctangent function executed by homodyne detectors, where sine and cosine signals are converted to phase, is that the phase can change a maximum of π radians between samples. So changes greater than π represent ambiguity as to whether the signal travelled a full clockwise or counterclockwise lap around the four quadrants [64]. Therefore, an absolute change higher than 180° between two adjacent milliliters may be a discontinuity caused by a new wrap cycle in the domain of 360°, influenced not only by the intrinsic nature of the system but also by noise that was still present in the measured data despite the stabilization mechanism. Under this condition, predictive algorithms were used to improve the π limitation [40] as performed by other publications about phase measurands from optical interferometers [30,64,65].

### 3.8. Three Actions of Phase Series Adjustments

Taking into consideration the aspects of phase detection reported in the earlier topic, three actions were applied on the phase data collected in this work.

Action (a)—Fixing Phase Discontinuities with 360° Rotations

In action (a), all absolute variations above 180° between two adjacent milliliters were rotated by 360°. For example, if a liquid volume point is −270° and the earlier point was +45°, then a 315° absolute variation is detected and, therefore, the −270° point is changed to +90° after a 360° rotation. If this new value also caused new further absolute variations above 180°, the respective liquid volumes were rotated as well.

Action (b)—Unwrapping the Phase Series

In action (b), the phase data series was unwrapped, in which the angles were rotated by ±180° based on the following thresholds vector: (−180° −90° −60° −30°). The minus signal of a threshold value works only on up–down variations, and the plus signal on down–up variations. The algorithm ran on each phase difference between two adjacent milliliters, and if it was higher than the threshold value and was in the signal direction of the threshold signal, the point and the remaining series were rotated by 180°. Up–down movements were rotated by +180° and down–up by −180°. In the end, these rotations performed an unwrap operation in the whole phase series, respecting the arctangent function properties as the earlier and the updated angles still kept the same tangent values. Different values of this vector were assessed, and it was found that positive thresholds (high down–up movements) did not affect the unwrapping process.

Action (c)—Excluding Phase Points with Deviations Above 60°

In action (c), as a strategy for phase noise treatment, a phase point was excluded when the standard deviation in the measurements group of liquid volumes was greater than 60° during data collection.

### 3.9. Fitting Model for Phase Characterization

After performing the three actions of phase adjustments, the best result was selected based on the greater R-squared (R^2^) and minor root mean square error (RMSE) according to the fitting model assessed for the final series of the unwrapped phase. Among the models evaluated, there were polynomials, exponentials, Fourier series, Gaussians, power series, and sums of sines.

RMSE and RMSE% are calculated by Equations (22) and (23), respectively, where $Y_{\mathrm{Ti}}$ and $Y_{\mathrm{Pi}}$ are the true and predicted values, respectively, of the ith term in the phase series composed by N terms.
(22)$$\mathrm{RMSE}=\sqrt{\frac{1}{N}\overset{N}{\underset{i=1}{\sum }}{\left(Y_{\mathrm{Ti}}-Y_{\mathrm{Pi}}\right)}^{2}},$$
(23)$$\mathrm{RMSE}\%=\sqrt{\frac{1}{N}\overset{N}{\underset{i=1}{\sum }}{\left(\frac{Y_{\mathrm{Ti}}-Y_{\mathrm{Pi}}}{Y_{\mathrm{Ti}}}\right)}^{2}}\times 100\%,$$

### 3.10. Machine Learning Algorithms

This work employed machine-learning algorithms under supervised learning to predict liquid volumes from amplitudes and phases of acoustic waves. Two distance-based techniques were tested, Rocchio and k-NN [66], with five distance metrics: Euclidean, Mahalanobis, cosine distance, cosine similarity, and Manhattan. Rocchio is a classifier based on the nearest centroid defined by the training data, and k-NN is based on the k-nearest neighbors to the training data.

In total, six scenarios were evaluated, and each scenario was defined by a specific choice of object tuples (lines) and attributes (columns) from the 11,220 × 64 matrix of collected data. A group of selected tuples defines a dataset, and the group of selected attributes defines a data model. In this case, a data model can include, for example, water temperature, so as to see the influence of this field in the results. Further, each scenario has its own criteria to separate the training from the testing datasets. As an example, a training dataset can be composed only of the twentieth object tuple of the liquid volume points (mode centroids), and the testing dataset by the remaining objects, or the training set can be composed of a random selection of 75% of all tuples, and the testing dataset includes the remaining 25%. In addition, other characteristics, such as distance metrics used and the “k” number of neighbors in k-NN were assessed.

When the training dataset is composed of one tuple standing for the centroid of a milliliter, both Rocchio and 1-NN reach the same accuracy. So, in these cases, the 3-NN is presented, while other cases used 1-NN as it delivered better accuracy. For all scenarios evaluated, both training and testing datasets were normalized to avoid distortions caused by the scales of the attributes. Each liquid volume point was defined as a class (label) with which a new aleatory sample of phase and amplitude was processed to obtain the respective liquid volume. The data were balanced, as the classes have the same number of samples, although they can vary a little with scenarios that excluded outliers.

### 3.11. Gaussian Process Regression

In addition to the use of classification algorithms, this work also assessed machine-learning regression versions based on the same fitting models used to characterize unwrapped phase data series. However, instead of testing only the mode values of amplitudes and phases, the regression models with machine-learning were applied to all database points with the same random selection of 75%/25% to define the training and the testing datasets, respectively. The best result in terms of R2 and RMSE was obtained from Gaussian process regression (GPR), whose values are presented in the result section of this paper. GPR assumes that the target variables are Gaussian distributions and makes use of covariance matrices derived from data attributes to predict a fitting model [66].

## 4. Results and Discussion

### 4.1. Sensitivity, Resolution, Bandwidth, and Other Technical Characteristics

Regarding acoustic bandwidth, the system was designed to operate at the fixed frequency of 100 kHz with the ultrasound transducer able to work at the maximum frequency of 1.5 MHz, and Figure 4 shows an experimental result of a 6 dB bandwidth at 700 kHz, ranging from around 30 kHz to 730 kHz, inside which specific frequencies of 80 kHz, 220 kHz, and 340 kHz presented high peak values in the response curve. This response was computed under a fixed liquid volume, as different volumes present different voltage amplitudes for a fixed frequency. The choice of the FOH system’s fixed operation frequency took into consideration experimental bandwidth, the Fresnel’s zone restrictions, and the longest possible acoustic wavelength in the water.

The EF loop circuit cutoff bandwidth was calculated following [55], and Figure 5 shows the oscilloscope graphical results used to obtain the sensitivity $K_{p}=\frac{31.5\pi }{4.24}=23.34$ rad/V and $K_{d}=\frac{1.06}{\pi /2}=0.67$ V/rad after driving the PZT with a 50 Hz and V_pp_ = 4.24 V triangular wave. The only requirement to choose the triangular wave’s frequency was to make it as small as possible in order to allow counting the number of cycles in the resulting sinus modulated wave. Therefore, the experimental high cutoff frequency of the EF circuit loop resulted in $f_{\mathrm{EF}}=\frac{23.34\times 0.67}{2\pi \times 100\times 10^{3}\times 4.7\times 10^{-6}}=5.33$ Hz, which was enough to mitigate most environmental noise. A replacement of capacitor C1 from 4.7 µF to 1 µF raised this value to 25.07 Hz, which is close to the value experimentally found by [55].

### 4.2. Original Amplitude and Phase Data Series Collected

The original collected values of amplitudes and phases per each liquid volume point are plotted in Figure 6. As a homodyne detector (phasemeter), each amplitude in volts stands for the optical phase displacement between the signals from the MI’s arms, related to a specific liquid volume, and each phase point is the electronic acoustic phase difference between the input and the output signals from the FOH system. Phasemeters recover the phases from the four-quadrant arctangent according to the ratio between two quadrature signals (a sine and a cosine). The relationship between the expected phase and the phase effectively measured should be linear, but there are often distortions caused by spurious effects in the optics or in the signal processing that result in the nonlinearities and periodic errors [35] seen in Figure 6.

Specific characteristics can be noted from the originally collected data. Both amplitude and phase series are nonlinear with liquid volumes; they are not intrinsically correlated to each other; there are different peaks and valleys when only S1 is under water than when both S1 and S2 are submerged; below S2, the phase varies around the limits of −45° to +45°, covering a range of 90° as limited by the quadrature; above S2, the phase measurand shows maximum and minimum values that touch the limits of −270° and +90°, covering a range of 360°; above S2, the amplitude window of peaks and valleys dislocates down; both series present limited linearity intervals with liquid volumes, as seen in Figure 7; the amplitude shows quadratic behavior at volumes of 820, 890, 920, 940, and 970 mL; the phase measures show discontinuity jumps near the volumes of 650, 690 to 710, 740, 860, 920, 940, and 990 mL; and the first measured phase at 440 mL presented a value smaller than −180°.

These results confirm nonlinearities, differences in using one or two optical coils as sensing elements, attenuation of acoustic fields as the volume of water increases, and that phase differences between the optical signals in the MI arms are not related to the phase differences between the acoustic signal input and output from the system.

### 4.3. Phase Measurand Analysis

Three actions were applied to the collected phase data in order to: (a) fix the discontinuities; (b) unwrap the data series; and (c) mitigate the noise.

In action (a), after all rotations, the new phase profile is shown in Figure 7, from which it is noted the effect of dislocating the whole data series from the limits of −270°/+90° to a new range of −180°/+180°.

For comparison with the original data presented before, Figure 8 shows the original amplitudes and the new phase series corrected with the 360° rotations.

A derivative analysis of both amplitude and phase measurands is presented in Figure 9, with common phase sensitivities reaching 20° degrees/mL (349 milliradian/mL) and amplitudes in 70 mV/mL. The dashed lines are one standard deviation over means for both measurands, separated by liquid volumes below and above the S2 coil. Comparing variations due to the influence of the S2 coil, the amplitude deviation decreased from 44 mV to 31 mV, while the phase increased from 24° to 37°.

That is, the presence of S2 decreased the sensitivity of amplitude (homodyne detection related to the optical phase differences between the signals inside the MI arms) but increased the sensitivity of the acoustic phases (related to the electrical phase differences between the signal input to the UT and the measurand signal delivered by PD3). Based on that result, future analysis would take S2 out of the system and verify if coil S1 alone could detect different series of amplitudes and phases, including their derivatives.

Figure 10 shows plain linear intervals, with numbers at the ends of straight lines being the first and final liquid volume of each interval. This figure also shows that the majority of phase points have their mode values (“x”) close to their mean values (“+”), which is a characteristic of normal distributions, although neither normal fitting test was performed over the data, as it was not a goal of this work.

Figure 9 shows that after executing action (a), the phase derivative series are mostly confined in the limits of −20°/20° with one standard deviation of 37°. So, a maximum admissible standard deviation error of 60° for a point in action (c) is a conservative value, although other values have been evaluated. So, action (c) caused the elimination of 4% of collected data, and Figure 11 shows the resulting series after each one of the three phase actions were executed. The bottom of Figure 11 also shows room and water temperatures, both measured during data collection days represented by small circles on the graph.

According to Figure 12, while the original phase derivatives reached a six-standard deviation of 290°, the final adjusted phase series ended with a six-sigma deviation of 82°. This result was achieved after the execution of three sequenced steps of phase adjustments, whose algorithms improved upon the π constraints of angle measurands [40]. In fact, those steps detected and cleaned phase jumpers between adjacent milliliters with absolute values higher than 180° and unwrapped the phase series after removing phase points with deviations equal to or higher than 60° from the original data.

Derivative analysis showed the improvements achieved by these proposed phase adjustment algorithms, which had the goal of finding a final phase characterization for the FOH system. Those actions were based on parameters that could be previously configured, allowing the MA to apply them as long as new data were collected online.

Finally, Figure 13 shows the phase characterization after fitting the unwrapped phase with a sum-of-sines of nine terms, resulting in R2 of 0.9994 and RMSE% of 4.15%. The model was configured to run with least absolute residual method and nonlinear least squares.

The sum-of-sines fitting function is expressed by:(24)$$L_{\mathrm{ml}}=\overset{9}{\underset{i=1}{\sum }}a_{i}*\sin\left(b_{i}*\Delta \emptyset_{s}+c_{i}\right),$$
where $L_{\mathrm{ml}}$ is the predicted liquid volume in milliliters, and $\Delta \emptyset_{s}$ is the angle detected and processed by the FOH system’s application after the three phase treatments are applied. The coefficients $a_{i}$, $b_{i},$ and $c_{i}$ of the adjusted function are listed in Table 1.

### 4.4. Amplitude Measurand Analysis

The amplitude and its derivative series have already been presented in the earlier section to allow their comparison with the phase series. So, a next analysis is to show the presence of linear intervals, as shown in Figure 14. This figure also shows that most amplitude points have mode values quasi-equal to mean values, what is also a characteristic of normal distributions.

Figure 15 shows the derivatives of amplitude to liquid volumes, with limited ranges of linear behavior.

The amplitude analysis shows that the collected data are nonlinear, and it is not obvious how to derive a math function to fit these data. In this sense, the next section presents an analysis using both phase and amplitude mode points.

### 4.5. Phase and Amplitude Analysis

This topic evaluates both amplitude and phase as coordinates of a 2D Cartesian space. Figure 16 uses the corrected phases previously presented in Figure 7, and Figure 17 uses the unwrapped phases presented in Figure 13. Both graphs have the phase measurand in the *x*-axis and the amplitude measurand in the *y*-axis, with point coordinates defined by their mode values. The integer numbers plotted over the sequenced points are the respective liquid volumes in milliliters. Considering the substantial number of measured points, not all values were plotted to preserve an adequate graphical visualization by the reader.

Figure 16 shows points standing for a linear sequence of liquid volumes, but with the evolute as curves of amplitude and phase coordinates making clockwise turns as long as the liquid volume increases linearly, forming patterns like “snail paths” that are overlapped by other paths, showing nonlinear behavior of the FOH system. This pattern also shows spaces with a high concentration of liquid volume points and empty spaces in the vicinity. There is also a difference in space point occupation when S1 is submerged (liquid volumes below S2) compared to when both S1 and S2 are underwater. In fact, the unwrapped phases are seen only with measurements above S2.

### 4.6. Rocchio and k-NN with Different Distance Metrics

This topic presents six scenarios of data analysis. Scenarios one to four used the phase adjusted to the range of −180°/+180°. Scenario five used the unwrapped phase, and scenario six used the original phase without any adjustments.

Figure 18 shows the results of first scenario that evaluated the mode values of phase and amplitude related to a liquid volume point. The mode values were used as the training dataset, and they were evaluated against themselves as testing data. In this scenario, an accuracy of 100% was expected, but the accuracy of 96.8% shows that 3.2% of the centroids overlap; that is, although they stand for different liquid volumes, they have the same coordinates of amplitude and phase. This result corroborates the behavior of Figure 16 in which the “snail paths” overlap each other more than one time. This also shows that Rocchio is not applicable if data are composed of centroids of mode values. Additionally, k-NN can be seen as useless with an accuracy of 31.2%. The small circles in the graphs supply a visual notion of the wrongly classified points.

This first Rocchio result shows a trade-off in which the need to round numbers to allow the use of mode values has the collateral effect of increasing the probability of centroids overlapping. So, the number of digits to the right of the decimal points is an input parameter to the system. The fewer digits, the higher probability of overlapping. Based on the results of this first scenario, new data models and new selections of training and testing data were defined to improve analysis.

A second scenario used the same data model as the first one, but with modes replaced by means, and Figure 19 shows an accuracy of 100% for Rocchio and 29.8% for 3-NN.

If both modes and means are used in the data model as a third scenario, the Rocchio and 3-NN accuracies are 100% and 31.4%, respectively. A partial conclusion here is that both training and testing datasets, besides the data models, still need to be improved. Figure 20 shows a fourth scenario in which the training data are the centroids of modes and means of phase and amplitude, and the testing data are all the remaining points collected by the experimental setup excluding the centroids and the interquartile outliers.

The fourth scenario is more data intensive than the third, as it used more tuples as testing data, resulting in an accuracy of 57.8% for Rocchio and 25.0% for 3-NN. Additionally, 5-NN’s accuracy is 15.8%. In this sense, it is possible to note that training datasets composed only of centroids are poor data models, as their low distances and overlapping points are not adequate for distance-based algorithms.

Therefore, a fifth scenario was defined, in which the data model was improved, and its attributes were composed of point values, means, and modes (intermediate and finals) for both amplitude and phase, complemented by the water temperature. The fifth scenario considered the unwrapped and characterized phase data presented in Figure 13.

The choice of object tuples (lines) was aleatory to define the training and the testing datasets with the respective proportion of 75%/25%. Initially, no outliers were excluded. The k-NN was defined to one nearest neighborhood (1-NN) as it presented better results than when using three or more neighbors for all distance metrics.

As a result, for Euclidean distance, 1-NN reached an accuracy of 91.6%, and other k-NNs with three or more neighbors presented lower accuracies (3-NN = 87.9%; 5-NN = 85.3%; 9-NN = 80.2%; 15-NN = 71.6%; and 20-NN = 64.0%), reinforcing that 1-NN is always better.

Figure 21 shows the resulting accuracies for different distance metrics used with Rocchio and k-NN, in which 1-NN with Manhattan distance reached an accuracy of 97.0%. The cosine similarity metrics resulted in accuracy of 0% (zero) for both algorithms and for all scenarios evaluated so far.

These distance-based analyses could have stopped with the fifth scenario, as an accuracy above 95% was found. However, perceiving the result of Manhattan’s distance metrics in the fifth scenario, a sixth and last scenario was proposed with the goal of testing new attribute models and data sources only using that metric. The data sources were differentiated according to the three phase adjustment actions previously presented. A phase series could be (i) original; (ii) adjusted to −180°/+180°; (iii) unwrapped; or (iv) unwrapped with exclusions of deviations above 60°.

Testing revealed the best accuracy found for the sixth scenario occurred when the data model attributes and data tuple selection were performed the same as for the fifth scenario, but with the original phases as collected from the experiment without any adjustments. Fine-tuning of the data model was also applied, in which different combinations of interquartiles and percentiles were evaluated using the fields of outlier indicators originally registered by the FOH’s MA application. As there are four variables, sixteen combinations of two outliers and two measurands were evaluated, and their results are presented in Table 2. When an outlier exclusion is applied, it is shown by an “x” in the table. Figure 22 shows the best graphic results derived from theses analyses.

Although Test 2 of Table 2 presented the greatest accuracy of 99.4% among the 16 tests, Figure 22 shows that Test 1 also has a high accuracy of 99.1% but a cleaner graph, in which the distances between the correctly and the wrongly predicted liquid volumes are shorter. This result shows that the simultaneous application of both outlier exclusion strategies, interquartile and percentile on both amplitude and phase series, on Test 1 eliminated more noisy values than on Test 2. Therefore, although Test 1 (99.1%) has a slightly lower accuracy than Test 2 (99.4%), Test 1 is preferable as a model as it has classification errors with smaller deviations around the right liquid volumes.

The use of distance-based machine learning algorithms was prominent in this work, as the results achieved accuracies of 99.4% for 1-NN and 97.3% for Rocchio, with both using Manhattan distance metrics. Values of 3-NN = 99.1% and 5-NN = 98.5% were calculated, proving that one-neighborhood models have better accuracy. The accuracies for Euclidean distance in sixth scenario were 88.5% for Rocchio and 98.0% for 1-NN.

Manhattan metrics represent the distance between two points measured along axes at right angles of 90° degrees, following a grid approach instead of the straight lines of Euclidean distance [67]. When comparing measured and reference signals, similarity methods based on Manhattan metrics can be used to minimize the distances between signals [68,69]. Besides the better accuracy, Manhattan metrics are more preferable than Euclidean in terms of resource consumption, as Euclidean requires square-root operations, and Manhattan needs only the absolute value of a subtraction [70].

### 4.7. Gaussian Process Regression

A Gaussian Process Regression (GPR) algorithm was configured with the method of rational quadratic type, and the resulting R^2^ of 1.00 and the RMSE of 0.21164 are shown in Figure 23. An error of 0.2 over supervised data collected with the steps of 1.0 mL shows a great fit of the GPR over the nature of physics involved in this experiment.

Therefore, ending the results and discussion, different machine learning algorithms were evaluated, in which two were of classification and one was of regression type. A model fit to the unwrapped phase was also shown. The classification algorithms were assessed considering five different distance metrics.

After all analyses, the best results are presented in Table 3.

## 5. Conclusions and Future Work

In this work, a fiber-optic hydrophone (FOH) based on Michelson’s interferometer (MI) is actively stabilized by an electronic feedback (EF) loop circuit to mitigate mechanical and thermal noise coming from the external environment. The FOH system is employed as a hardware and software application to perform liquid volume measurements of a 1000 mL glass graduated cylinder filled with water with volumes ranging from 440 mL to 1000 mL. This range limitation is due to the setup and positioning of the ultrasound source inside the cylinder. The sensing elements are composed of two optical fiber coils placed at the sensor head and a third optical fiber coil placed at the MI’s reference arm. The coils are made of readily available and common SMF fiber. The reference coil is wound around a piezoelectric actuator to stabilize the system at the quadrature point of π/2, and the two sensor coils are submerged in water to detect acoustic wave patterns formed by a particular liquid volume, supplying distinct values of amplitudes (volts) and phases (degrees) as output measurands. Finally, due to the nonlinear behavior of theses measurands, machine-learning algorithms are employed under supervised learning to predict the liquid volumes.

The accuracy of 99.4% achieved by this work is a prominent result in liquid volume predictions for a system that employs acoustic detection through optical fiber sensors. Further, an RMSE of 0.21164 mL shows the feasibility of the FOH system. For the test tank used, the value of 0.21 mL is equivalent to 0.0714 mm of error in the height of the water column. An added contribution of this work is the characterization of the unwrapped phase measurand, in which a sum-of-sines fitting function reached an R^2^ of 0.9994 and an RMSE% of 4.15%.

Another contribution of this work is a practical demonstration of the homodyne demodulation approach working with an active stabilization mechanism over a Michelson’s interferometer, as [40] mentions that it is a challenge to stabilize such systems for sensing applications, and it is useful for laboratory analysis. The advantages and drawbacks of quadrature stabilization methods are related to the resolution of the interferometer, which is the smallest physical quantity that a sensor can measure, defined by the noise of the measurand. Multiple reflections in optical arms improve the resolution of a homodyne interferometer but can induce phase jumps and loss of laser coherence. Therefore, a compromise must be found between increasing resolution and losing coherence [35].

As a limitation, this first version of the FOH system was used to measured only one type of liquid (water), although the two optical coils in MI’s sensor head were intentionally constructed to find different fluids around them (e.g., water and oil), which will be done in future tests. Another limitation is the medium time of three seconds to collect and process one data sample. As a milliliter requires twenty samples, the FOH system takes a minute to predict liquid volume.

As suggestions, the system can be improved to detect different fluids and multifluid interfaces. This stabilized FOH can also be used with a few adaptations to perform ultrasound piezoelectric transducer calibration [18,58]. Third, the FBG structure purposed by [23,71] could include an optical coil inside the polymer diaphragms as an additional sensing mechanism to detect vibration patterns of liquids inside a production vessel. Finally, a long short-term memory (LSTM) neural network could be considered as a prediction model, as its recursive behavior would be capable of detecting a sequence of points in the “snail paths” shown in the 2D plane of modal amplitude and phase patterns.

## Figures and Tables

**Figure 1 sensors-22-04404-f001:**
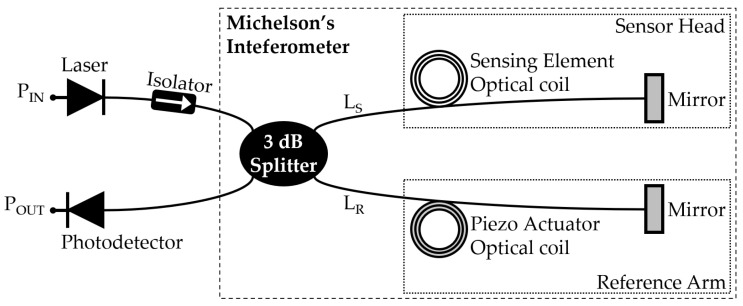
An FOH based on a Michelson’s interferometer.

**Figure 2 sensors-22-04404-f002:**
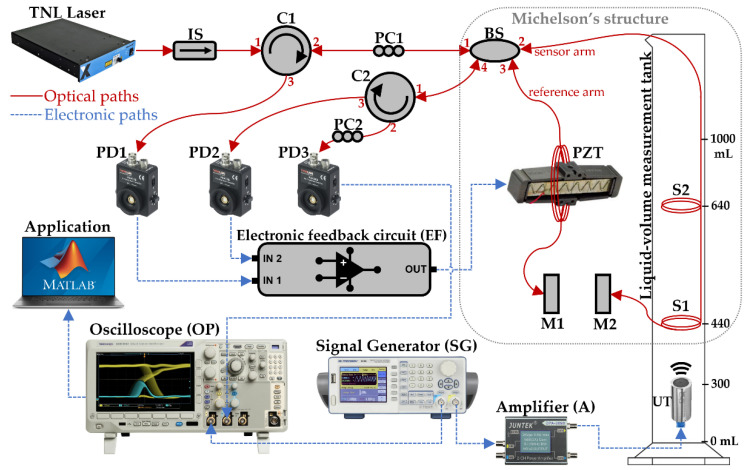
Fiber-optic hydrophone (FOH) system setup. TNL Laser at 1550 nm; IS—optical isolator; C1 and C2—optical circulators; BS—beam splitter; PC1 and PC2—polarization controllers; PD1, PD2 and PD3—photodetectors; M1 and M2—Faraday’s mirrors; PZT—piezoelectric actuator wound by an optical fiber coil; S1 and S2—optical fiber coil sensors; UT—ultrasound transducer; EF—electronic feedback loop circuit; OP—oscilloscope; SG—signal generator; A—3 dB amplifier; Application is software developed in MATLAB; liquid volume measurement tank.

**Figure 3 sensors-22-04404-f003:**
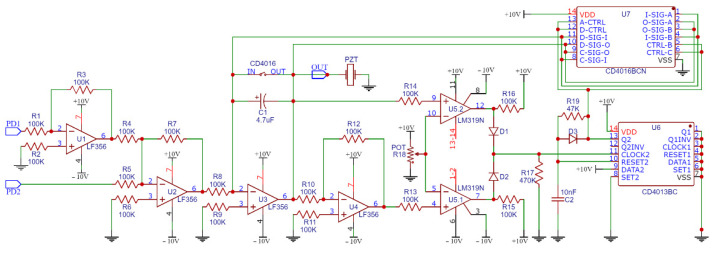
Schematic of electronic feedback (EF) loop circuit for Michelson’s stabilization.

**Figure 4 sensors-22-04404-f004:**
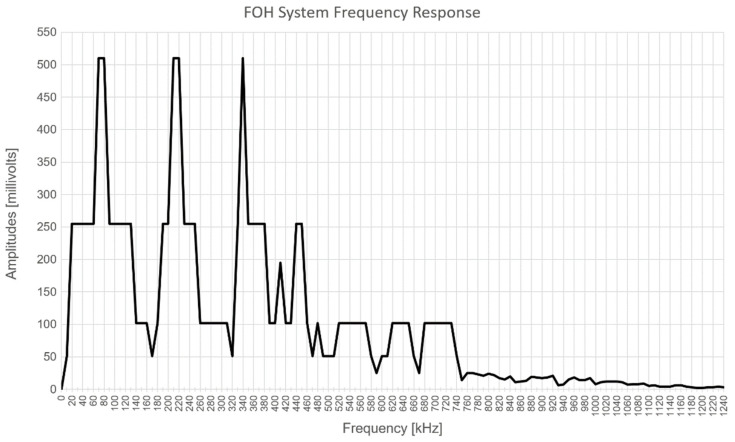
Experimental FOH system’s ultrasound frequency response curve.

**Figure 5 sensors-22-04404-f005:**
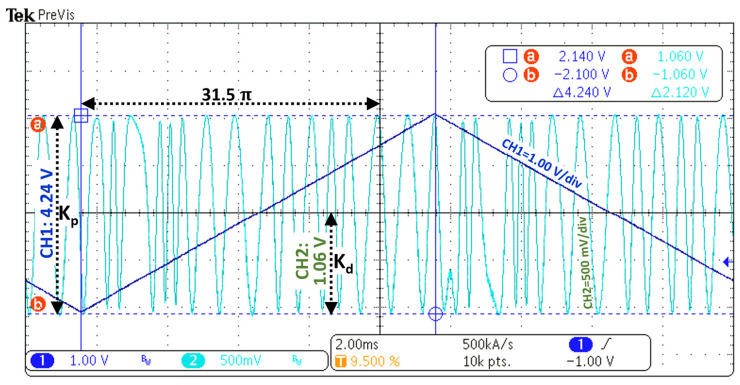
Coefficients $K_{p}$ (PZT with optical coil) and $K_{d}$ (photodiodes) experimentally obtained.

**Figure 6 sensors-22-04404-f006:**
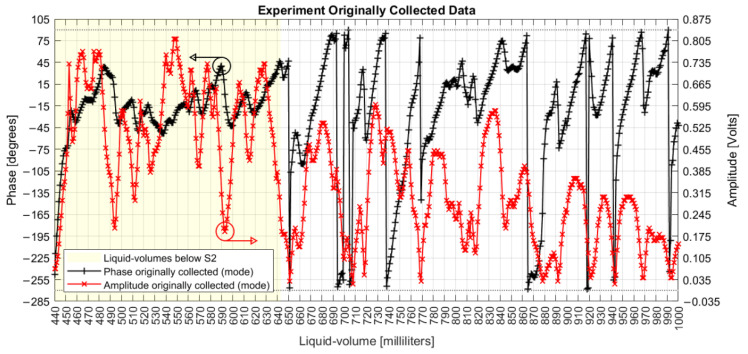
Original amplitude and phase mode values collected per milliliter.

**Figure 7 sensors-22-04404-f007:**
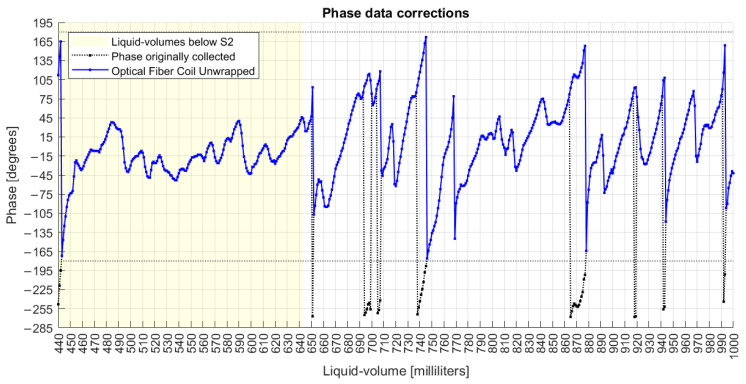
Phase measurand with correction of discontinuity jumps ≥180°.

**Figure 8 sensors-22-04404-f008:**
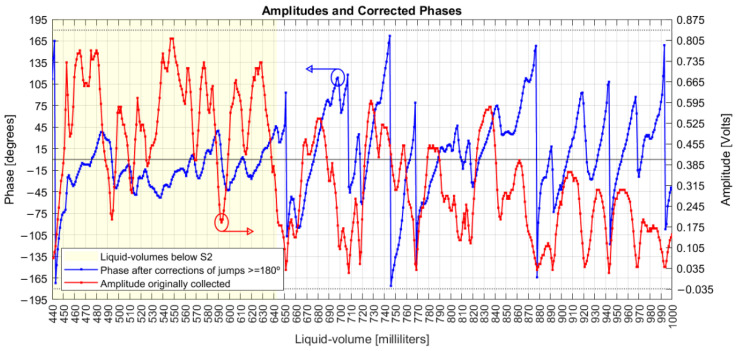
Original amplitudes and corrected phases.

**Figure 9 sensors-22-04404-f009:**
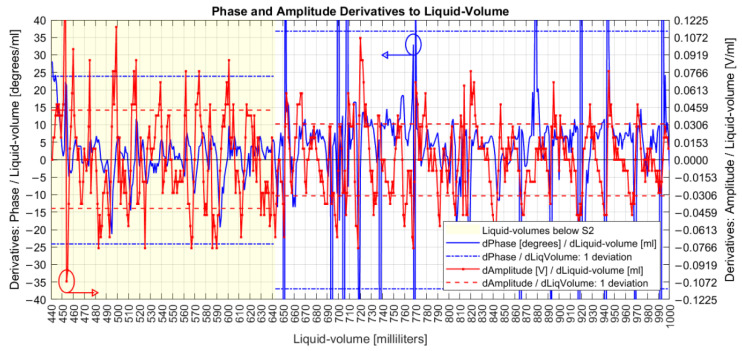
Derivatives of amplitudes and corrected phases.

**Figure 10 sensors-22-04404-f010:**
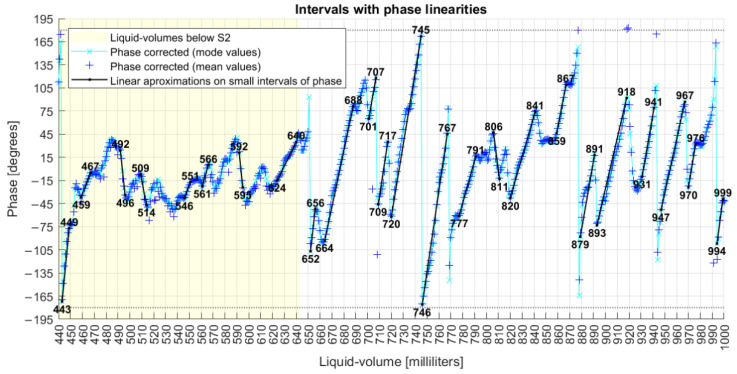
Liquid volume intervals with linearities with phase.

**Figure 11 sensors-22-04404-f011:**
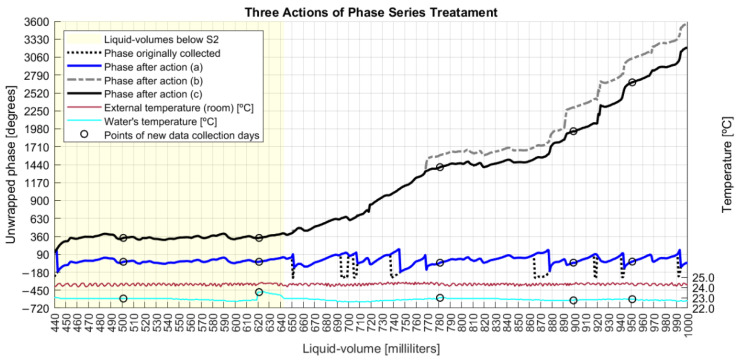
Three actions of phase series adjustments.

**Figure 12 sensors-22-04404-f012:**
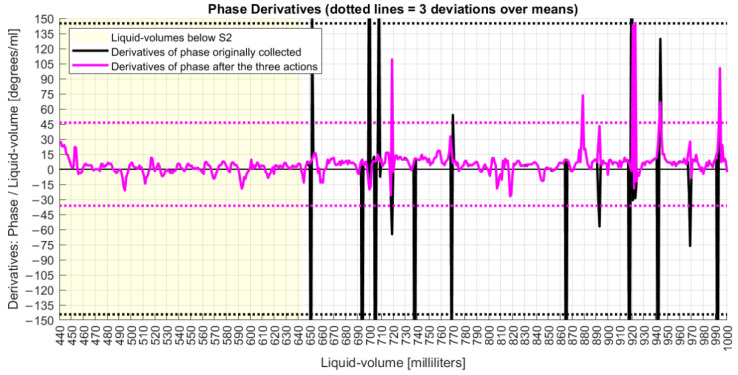
Derivative analyses of phase signals before and after corrections.

**Figure 13 sensors-22-04404-f013:**
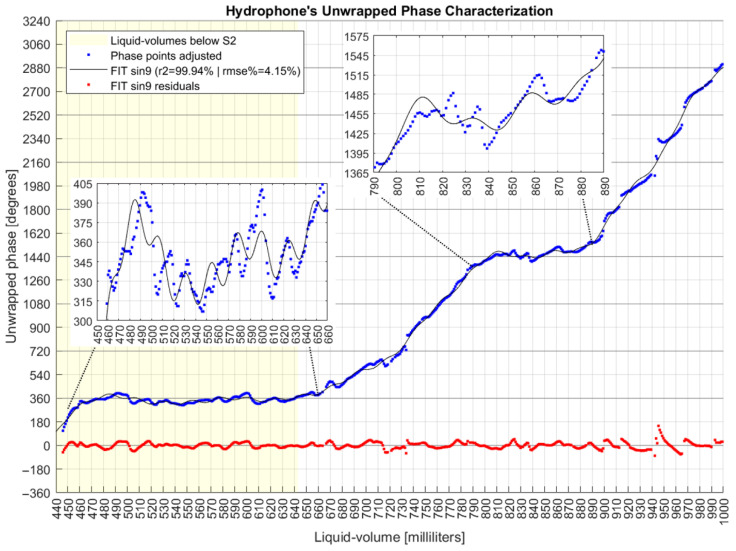
Hydrophone system’s unwrapped phase characterization.

**Figure 14 sensors-22-04404-f014:**
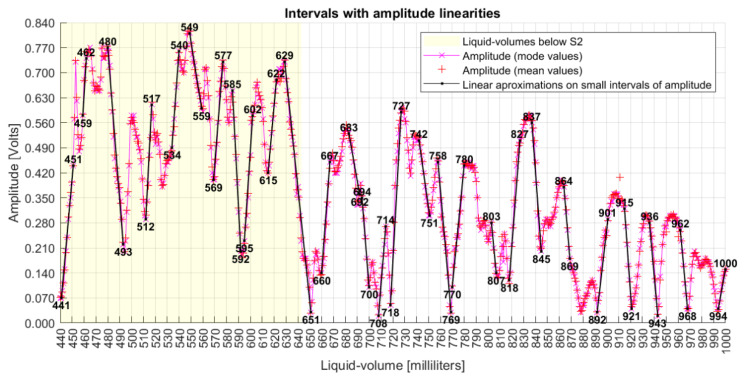
Liquid volumes with linear intervals on the amplitude series.

**Figure 15 sensors-22-04404-f015:**
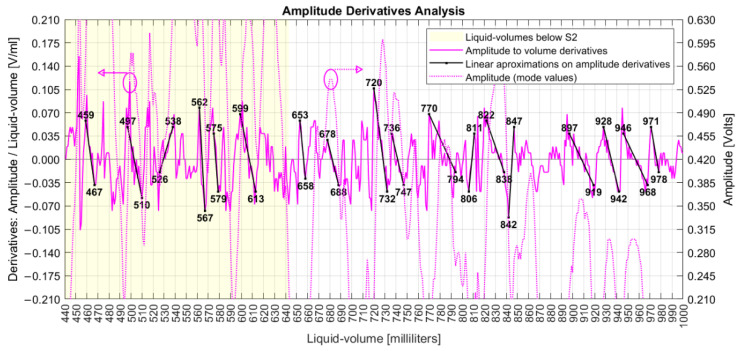
Liquid volumes with linear intervals on the derivative of amplitude series.

**Figure 16 sensors-22-04404-f016:**
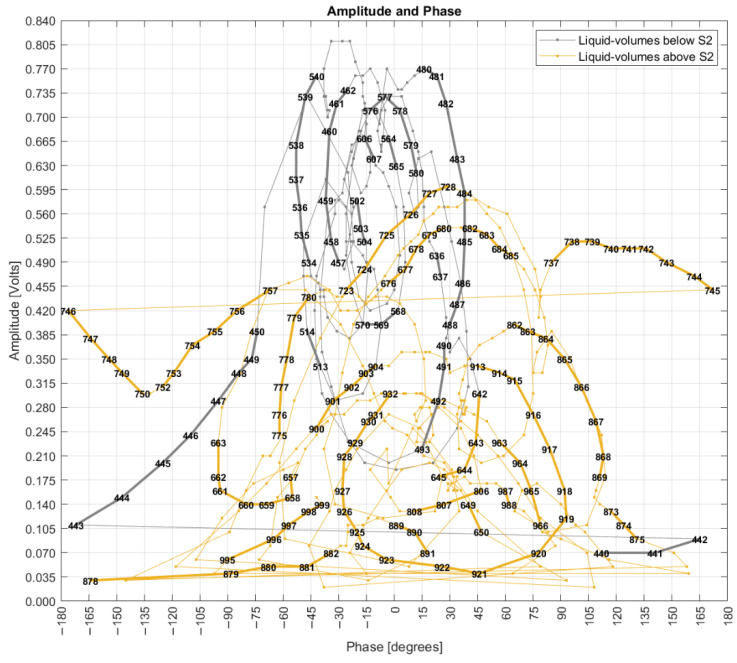
The 2D grade of amplitudes and corrected phases (only action “a” applied).

**Figure 17 sensors-22-04404-f017:**
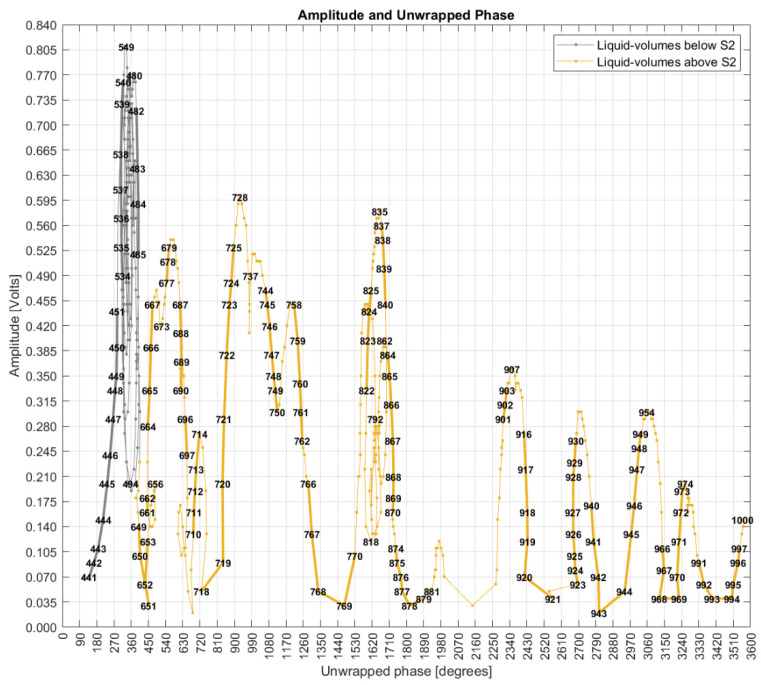
The 2D grade of amplitudes and unwrapped phases (after action “c” is applied).

**Figure 18 sensors-22-04404-f018:**
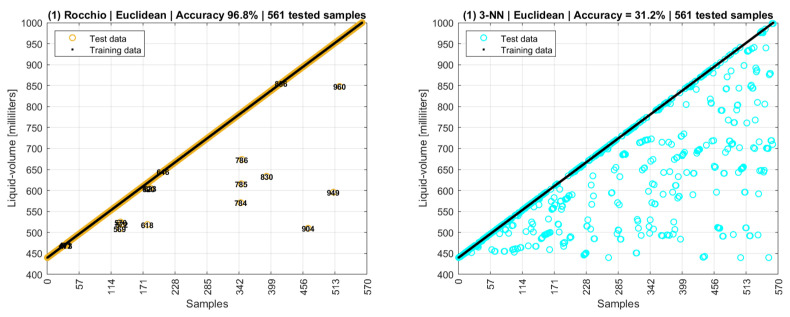
Rocchio and k-NN accuracies with centroids of mode values.

**Figure 19 sensors-22-04404-f019:**
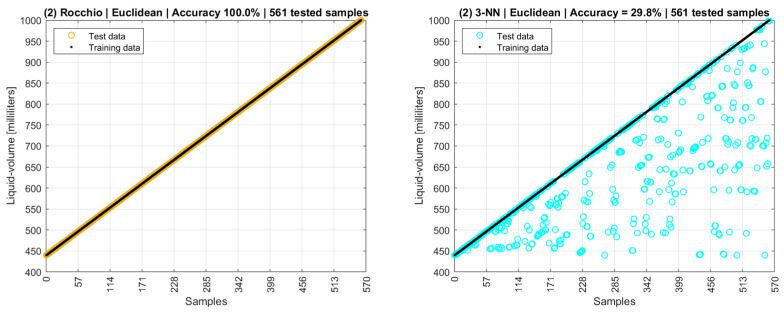
Rocchio and k-NN accuracies with centroids of mean values.

**Figure 20 sensors-22-04404-f020:**
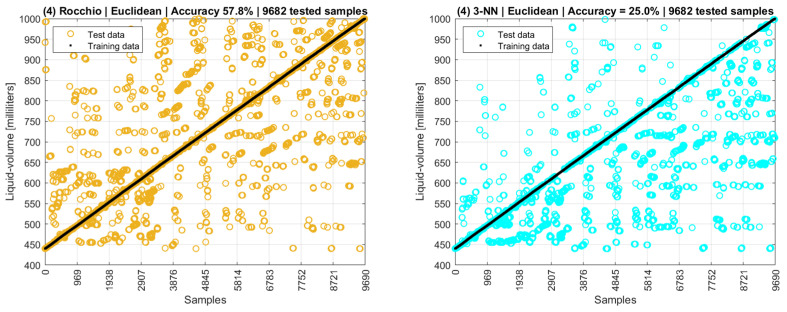
Rocchio and k-NN accuracies with complete datasets excluding outliers.

**Figure 21 sensors-22-04404-f021:**
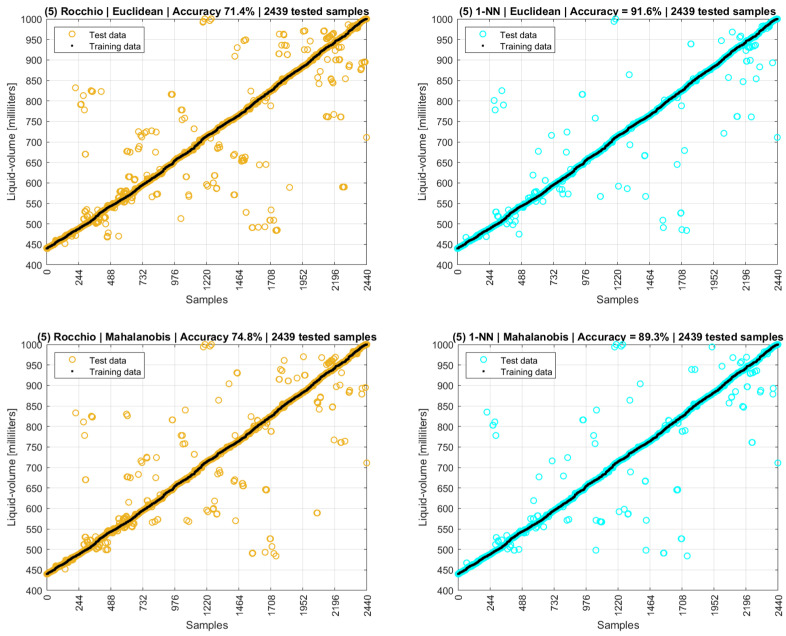
Fifth scenario. This figure has ten graphs distributed in two columns and five lines. Graphs on the left column are Rocchio and on the right are 1-NN. Each line stands for a different distance metric. Each graph title has the name of the distance used, the resulting accuracy, and number of tested samples.

**Figure 22 sensors-22-04404-f022:**
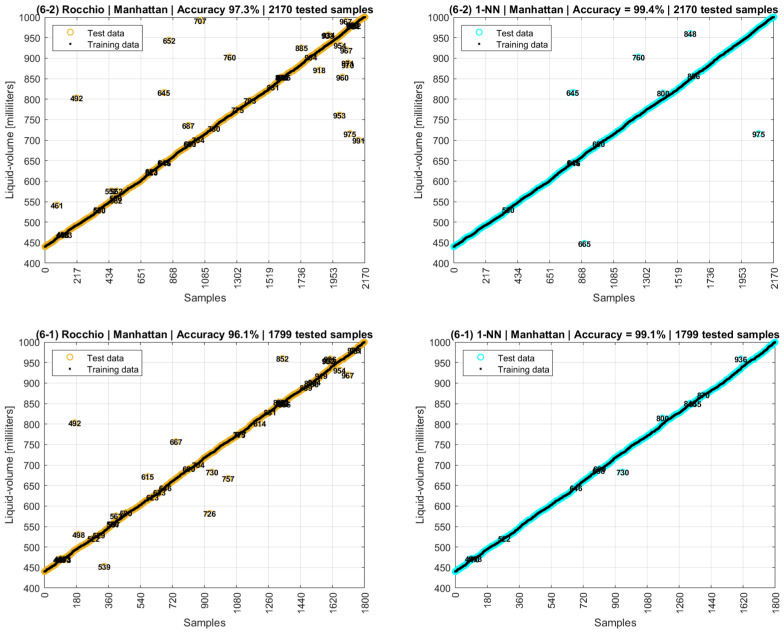
Rocchio and 1-NN accuracies with Manhattan distances over original data collected.

**Figure 23 sensors-22-04404-f023:**
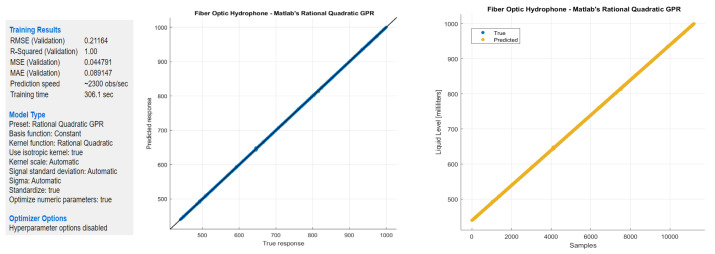
Gaussian process regression (rational quadratic) for the FOH system.

**Table 1 sensors-22-04404-t001:** Coefficients of sum-of-sines fitting function.

i	$a_{i}$	$b_{i}$	$c_{i}$
1	2419	0.7729	0.7455
2	1087	1.66	−2.039
3	288.9	3.401	0.8193
4	102.6	5.768	−0.8964
5	29.65	9.122	−2.307
6	5518	18.81	−1.821
7	5517	18.81	1.317
8	−4.334	13.82	9.008
9	13.23	45.2	−0.7065

**Table 2 sensors-22-04404-t002:** Sixth scenario results for Rocchio and 1-NN with Manhattan and outlier selections.

	Amplitude		Phase			Manhattan	
Test	Quartile	Percentile	Quartile	Percentile	Samples	Rocchio	1-NN
1	x	x	x	x	1799	96.1%	99.1%
2	x	x	x		2170	**97.3%**	**99.4%**
3	x	x		x	1801	96.3%	98.7%
4	x	x			2226	96.0%	99.3%
5	x		x	x	2120	96.9%	99.1%
6	x		x		2551	96.7%	99.0%
7	x			x	2122	96.3%	99.2%
8	x				2622	95.9%	99.2%
9		x	x	x	1811	96.3%	99.0%
10		x	x		2186	96.1%	98.9%
11		x		x	1814	95.9%	99.0%
12		x			2244	96.1%	98.8%
13			x	x	2240	96.1%	98.8%
14			x		2697	95.9%	98.5%
15				x	2244	95.8%	98.6%
16					2805	92.3%	98.0%

**Table 3 sensors-22-04404-t003:** Results of machine learning algorithms used for liquid volume predictions.

Machine Learning Algorithms		Distance/Type	Metrics	Value
Classification	**k-NN 1**	**Manhattan**	Accuracy	**99.4%**
	Rocchio	Manhattan		97.3%
	k-NN 1	Euclidean		93.1%
	Rocchio	Euclidean		75.1%
	k-NN 1	Mahalanobis		90.6%
	Rocchio	Mahalanobis		76.1%
	k-NN 1	Cosine Distance		90.9%
	Rocchio	Cosine Distance		64.6%
Regression	Sum of Sines	Nine terms	RMSE%	4.15%
			R^2^	0.9994
	Gaussian Process	Rational Quadratic	RMSE	**0.21164 mL**
			R^2^	**1.00**

## Data Availability

Data and software are available by reasonable request.

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
