# Peer review of "Fiber-Optic Hydrophone Based on Michelson’s Interferometer with Active Stabilization for Liquid Volume Measurement"

_sensors, 2022, doi:10.3390/s22124404_

Round 1
Reviewer 1 Report
This article reports on Michelson's interferometer-based optical fiber hydrophone to measure the volume of liquid. This research has clear objectives and goals. In my personal opinion, the paper is enjoyable to read and the presented graphs are informative. Therefore, I do recommend the manuscript to be published in the Sensors Journal.
Author Response
Dear Reviewer 1,
Please see the attachment.

Reviewer 2 Report
The authors demonstrate fiber-optic hydrophone based on MI with active stabilization for liquid volume measurement. The methods of fiber-optic sensing measurement, electric feedback, and machine learning are used to characterize and analyze the liquid volume measurement. The paper has some interest for the field of fiber-optic sensing. The manuscript can be accepted after responding the following questions.
1) How about the sensitivity, resolution and detection bandwidth of this fiber-optic sensor for the ultrasound detection?
2) What’s the advantage and novelty of this proposal? Fiber-optic ultrasound sensors base on structures of Fabry-Perot, Mach-Zehnder, Michelson’s, and Sagnac, have been deeply studied and applied in various applications.
3) In addition, the electric feedback loop circuit for the active stabilization is also refered from some references, not a new method.
4) I suggest adding at the Authors’ reference of the recent references about high-sensitivity fiber sensors for ultrasound detection of “Optics Letters 44, 3606 (2019); Optics Letters, 45, 1128 (2020); Optics Letters, 45, 3889 (2020).
5) English in the whole manuscript needs some revisions, such as line 87 “comparted to”, line 115 “and introduction”.
Author Response
Dear Reviewer 2,
Please see the attachment.

Reviewer 3 Report
The authors have presented the development of a Fiber Optic Sensor that can be used for pipeline liquid flow monitoring. The authors present both the hardware system for the signal acquisition as well as the signal processing for the detection algorithm. The work can be described as an incremental upgrade of previous similar sensing systems and is interesting to read and scientifically sound.
However, in order to ameliorate the readability and clarify some critical points in the scientific methodology the following changes are suggested to be undertaken by the authors:
1) In section 3 (Materials and Methods) the authors need to specificy the functionality of PD3. As far as I understood it receives light reflected from PD2 which is already polarized (twice?). What is the role of PD2 in this case? Since PD2 is the PD fed into the oscilloscope, and PD1 and PD3 compensate for various noise in the system it is important to briefly describe how (and why) there is a polarization control in PD3 and how the authors balance for the much different reflectivity levels between PD1 and PD3. This is an important point to be clarified as the noise cancellation system is central to the addition of noise in the system.
2) In lines 598-600 the authors need to expand on this part as it is not clear how the phase adjustments led to improvements.
3) The authors must be very careful with the use of word 'sensible' instead of 'sensitive'. in many parts of the text the word 'sensible' in use in lieu of 'sensitive'. I would highly recommend a thorough English proofreading before submitting the final draft.
The above points are strongly recommended to be addressed before this work goes into publication.
Author Response
Dear Reviewer 3,
Please see the attachment.

Round 2
Reviewer 2 Report
Thank the authors to respond my comments. I suggest this manuscript to be published in Sensors journal.